# Self-Supervised Joint Learning for pCLE Image Denoising [note 1]

**DOI:** 10.3390/s24092853

**Published:** 2024-04-30

**Authors:** Kun Yang, Haojie Zhang, Yufei Qiu, Tong Zhai, Zhiguo Zhang

**Affiliations:** State Key Lab of Information Photonics and Optical Communications, Beijing University of Posts and Telecommunications (BUPT), Beijing 100876, China; ykk@bupt.edu.cn (K.Y.); qiuyufei@bupt.edu.cn (Y.Q.); zhai_tong@bupt.edu.cn (T.Z.); zhangzhiguo@bupt.edu.cn (Z.Z.)

**Keywords:** probe confocal laser endomicroscopy, confocal, image denoising, self-supervised

## Abstract

Probe-based confocal laser endoscopy (pCLE) has emerged as a powerful tool for disease diagnosis, yet it faces challenges such as the formation of hexagonal patterns in images due to the inherent characteristics of fiber bundles. Recent advancements in deep learning offer promise in image denoising, but the acquisition of clean-noisy image pairs for training networks across all potential scenarios can be prohibitively costly. Few studies have explored training denoising networks on such pairs. Here, we propose an innovative self-supervised denoising method. Our approach integrates noise prediction networks, image quality assessment networks, and denoising networks in a collaborative, jointly trained manner. Compared to prior self-supervised denoising methods, our approach yields superior results on pCLE images and fluorescence microscopy images. In summary, our novel self-supervised denoising technique enhances image quality in pCLE diagnosis by leveraging the synergy of noise prediction, image quality assessment, and denoising networks, surpassing previous methods on both pCLE and fluorescence microscopy images.

## 1. Introduction

Probe-based confocal laser endomicroscopy (pCLE) is a technology that applies microscopic imaging techniques to endoscopic examinations in the medical field [1]. Optical scanning confocal microscopy is a mature technology that offers a significant advantage over traditional microscopic imaging, allowing for selective reduction of light from out-of-focus planes, thereby enabling depth imaging [2]. Clinical trials have demonstrated the value of this new technology. When utilizing pCLE, a laser beam is employed to illuminate the area of interest within biological tissues [3]. Currently, pCLE has found application across a wide range of biomedical scenarios, including the cavity structures of the digestive system, respiratory system, and urogenital system tissues or organs [4,5].

Due to the utilization of fiber bundles (FB) in pCLE, the spatial resolution of the FB imaging system is constrained by the core diameter and density of the optical fibers [6]. Additionally, variations in the light transmission properties of individual fibers and their surrounding sheaths result in the formation of honeycomb patterns, which superimpose on the imaging results and hinder the precise analysis of objects [7]. Over the past few years, various methods have been developed to eliminate honeycomb patterns, such as employing bandpass filters in the Fourier domain [8,9,10,11]. However, selecting an appropriate threshold in the frequency domain to eliminate honeycomb patterns without blurring the underlying imaging structures poses a significant challenge, which would severely affect image clarity [12,13,14]. Therefore, there is an urgent need to remove noise and enhance resolution.

In traditional deep learning, denoising methods typically involve neural networks learning the mapping relationship from noisy images to clean images [15]. This is a supervised learning approach, and to apply supervised learning within deep neural networks, a sufficient amount of labeled data is required [16,17,18,19]. However, this method faces some bottlenecks in practical applications, particularly in the medical field, where obtaining true clean images for training data can be extremely challenging. The noise value of a noisy pixel not only exhibits spatial correlation with the surrounding noise but also demonstrates some correlation between different color channels within a single pixel [20]. Existing techniques mostly focus on removing Gaussian noise, while most medical images are functional images where factors such as a single light source, detection method, and human body thickness often lead to non-uniform noise distribution [21]. For example, the noise in pCLE images may vary with factors like different cameras and gastrointestinal environments making traditional deep learning denoising methods potentially limited in dealing with situations where obtaining true clean data is challenging as in medical images.

To address this challenge, many methods have incorporated noise-based prior information to handle degraded images [22,23,24,25,26]. One advantage of this approach is that it can handle specific levels of noise variance, as the network has encountered this type of noise during training [27]. However, they often struggle to generalize well to unseen noise levels or types. In the development of self-supervised denoising methods, several strategies have been proposed to train them exclusively on noisy images [28]. Krull introduced the first self-supervised method called Noise2Void [29], which utilizes the concept of blind spots to avoid the network learning a constant mapping from noisy images. The idea is to avoid this by randomly sampling pairs of masked and unmasked regions within the same image, represented as $\left(Y_{i}^{masked},Y_{i}\right)$. However, the mask of blind spot network leads to the loss of high-frequency information, and this method assumes noise independence, making it less effective for structured noise. Batson [30] demonstrated that different masks can also influence denoising performance, and they replaced the mask with the average of surrounding pixels. Another approach, as presented in [31], involves designing masks based on masked pixels and adding a noise prediction network trained jointly with the denoising network to improve denoising performance. Recent research DIP [32] observed that the structure of the generator network is sufficient to capture image statistics before any learning, allowing it to be trained to recover images from reconstructions without requiring extensive datasets. This was the first study to directly investigate the priors captured by deep convolutional generative networks, rather than learning network parameters from images. Building on this research, ref. [33] demonstrated that a single noisy image itself can be used to train denoising networks with competitive performance. The results of single-image self-supervised learning not only provide a practical neural network-based image denoiser but also inspire further research into self-supervised learning in other image restoration problems [34,35]. Moreover, noise prediction plays an important role in self-supervised denoising, and the CDB-Net method of Feng et al. has shown good results in the field of noise prediction [36]. However, inherent limitations persist in the efficacy of information extraction from such predictions [37].

While utilizing noise prior information can offer effective solutions, there is a need for further research to enhance the method’s generalization performance to accommodate unknown noise levels and types. To address this issue, we introduce an evaluation criterion throughout the entire training framework to utilize metrics for determining which image exhibits the best restoration quality. We propose a no-reference image quality assessment metric, and subsequently we trained a neural network to select the optimal denoised image during training. This assessment metric, combined with state-of-the-art denoising techniques, has been applied to denoise pCLE images and other fluorescence microscopy images lacking clear reference images. Our approach achieved significant improvements in denoising tasks for pCLE images and fluorescence microscopy datasets.

## 2. Materials and Methods

### 2.1. Architecture

The self-supervised approach can be considered as a specialized form of unsupervised learning that mimics supervised learning through self-imposed tasks, rather than relying on predetermined prior knowledge. In contrast to fully unsupervised settings, self-supervised learning constructs pseudo-labels using information inherent in the dataset. The automatic acquisition of these pseudo-labels is crucial in self-supervised learning. In this study, we propose a self-supervised-constrained pCLE noise image denoising network.

This network consists of three scale branches, as illustrated in Figure 1. The overall architecture includes a non-blind denoising sub-network D-Net, a quality assessment sub-network Q-Net, and a noise prediction sub-network N-Net. In our previous work, we introduced a self-supervised learning method that incorporates an image quality assessment network to identify the optimal moments during training.

The deep learning method for solving the image denoising problem can be transformed into training the function $f_{\theta }$ according to the unknown parameter θ. In order to prevent the image from learning a constant mapping to a noisy image, the image is processed before the denoising network with an added mask, and the training samples and the learning objects are the images after the added mask, respectively; they can be expressed as $\underset{\theta }{\min}\sum_{m}L(f_{\theta }(\hat{y},y-\hat{y}))$. Since the self-supervised denoising algorithm we use has no clean data and uses only noisy images and masks, the goal is to minimize the self-supervised loss of the form $\theta \to \sum_{i=1}^{N}{\left\|f_{\theta }(Y_{i}^{masked}-Y_{i})\right\|}_{2}^{2}$, where $Y_{masked}$ is an image in which the pixel $Y_{i}$ has been masked using M. Thus, for the image denoising network, the only a priori information about the network is currently the structure of the network itself, so N-Net is used to predict the image noise information as new image features to be trained along with the noisy image.

N-Net extracts noisy observations y to generate an estimated noise level map $\left(\left(\sigma \left(y\right)=F_{E}\left(y;W_{E}\right)\right)\right)$, where $W_{E}$ represents the network parameters of N-Net. The output of N-Net is the noise level map, as it has the same dimensions as the input y and can be estimated through a fully convolutional network. Then, D-Net takes both y and $\sigma \left(y\right)$ as input and produces the final denoised result $\left(\left(x=F_{D}\left(y,\sigma \left(y\right);W_{D}\right)\right)\right)$. The noise network and the noise prediction network share the same parameters. The role of D-Net is to perform denoising under the guidance of noise intensity. N-Net’s extracted noise information and noisy image are input into the denoising network D-Net. During the training process, as the number of iterations increases, the image tends to be reconstructed.

We use an advanced Quality Assessment Network called Q-Net to evaluate reconstructed images. This network assesses the quality of reconstructed images, generating a score trend map for each image. By using the results from Q-Net, we figure out the best restoration moment for each image. It is important to note that detailed explanations of both D-Net and Q-Net can be found in previous works. Therefore, this paper mainly focuses on providing a comprehensive and detailed explanation of the N-Net network.

### 2.2. Architecture of the Noise Prediction Network

To overcome the limitations of existing techniques, this invention leverages Convolutional Neural Networks with the aim of providing a more precise prediction of the variance of real noise and using this estimated noise variance to assist in image denoising. The technical approach employed in this invention is based on a deep learning method that combines noise prediction and image denoising. For a given noisy image, the image model is defined as y=x+ε. Here, x represents the noisy image, y represents the original image, and ε represents the noise introduced during the camera’s processing. This approach integrates noise prediction and image denoising into a joint framework, utilizing deep learning methods to enhance the quality of denoised images by accurately estimating the characteristics of the noise present in the captured images. N-Net is a fully convolutional network consisting of 20 convolutional layers and does not include pooling layers or batch normalization operations. Each convolutional layer, except for the output layer, is accompanied by a non-linear activation function. Its architecture is depicted in Figure 2.

D-Net utilizes a U-shaped network framework, where the encoder employs max-pooling operations for downsampling, aiding in the exploration of multi-scale information within the image and expanding the feature receptive field. The decoder uses bilinear interpolation for upsampling, facilitating the reconstruction of denoised images. This setup ensures that both N-Net and D-Net are well-structured to perform their respective tasks in the image denoising process.

## 3. Results

### 3.1. Dataset

In the case of pCLE images where actual data is not available, our dataset is solely used for evaluating the visual quality of denoising. Thus, we train and evaluate our method on the Widefield2SIM (W2S) dataset, which is an openly accessible microscopy image dataset. W2S is one of the representative real-world datasets, containing well-aligned noisy-clean image pairs for training. We selected this dataset because we aimed to facilitate a fair comparison, utilizing the same dataset as used by the authors in [31,38].

In our experiments, W2S-1, W2S-2, and W2S-3 represent three channels of the W2S dataset [39], each with 120 fields of view (FOV), which can be treated as individual datasets. For each FOV, there are 400 noisy images with size of 512 × 512 pixels, and only one observation is used for training and evaluation. Across all three datasets, we generate the ground truth images using image averaging. The acquisition of noise-free images involves capturing 400 consecutive shots of a static scene and then taking the average. The noise labels are obtained by taking the difference between the noisy images and the noise-free images. Noise variance maps reflect the fluctuations across the 400 images with respect to the noise-free image. The standard deviation is computed from the variance map, which is then normalized.

### 3.2. Implementation

#### 3.2.1. D-Net

In our experiments, we use the network parameters: A 3-depth U-Net with 1 input channel and 64 channels in the first layer; we use padding to the 3 × 3 convolution layer, meaning that we do not change the height and width of the feature layers [40]. Since the feature layer size is not changed after convolution, the two feature maps can be directly stitched together, so there is no need for center cropping, and the final obtained convolution layer height and width are kept the same as the input convolution layer height and width. The number of iterations is set to 3000, and the image is saved every 100 iterations.

#### 3.2.2. Q-Net

Our implementation is based on the hyperIQA implementation [41,42,43]. Q-Net includes three parts of functions. The first part is the semantic extraction part of the framework based on ResNet50. The second part is the perceptual rule-building part; the input is the output of the last layer of ResNet of feature extraction. The third part of the quality prediction part of the network structure is relatively simple, with only four FC layers, but the parameter weights of these four FC layers are provided by the second part; the final output is the final score of the image.

#### 3.2.3. N-Net

To set network parameters, we specify that the convolutional kernels are all of size 3 × 3, and each convolutional layer has 64 kernels. To ensure that the image size remains unchanged after each convolution and to prevent boundary effects, we apply zero-padding during each convolution. The initial learning rate is set to 1 × 10^−4^ (0.0001). We employ the Adam optimization algorithm.

### 3.3. Evaluation on the Image Denoising Network

In order to test the performance of the denoising network and determine if it is consistent with the denoising process, where the images are first recovered and then noise is applied, we use the W2S dataset for image recovery and save the images within the number of iterations in order to calculate the PSNR and verify whether it is consistent with the trend of all getting better before getting worse in the recovery process.

Figure 3 shows the method of recovering a single image and the change of the image during the recovery process. It can be seen that the image gradually becomes clearer during the reconstruction process, but the trend of image quality change is not linear, and according to Figure 3, we can initially speculate that the image quality becomes better and then worse within a certain number of iterations.

In order to further prove this point, the change trend graph is obtained as in Figure 4 by calculating the PSNR during the recovery process of the two datasets in the W2S dataset. It can be seen that there is still no upward trend in the two images after the number of iterations has been extended to 2800 times, and before that, the PSNR becomes higher and then lower, and the number of iterations is not the same for different datasets to reach the peak, based on which it can be concluded that the necessity of adding Q-Net.

### 3.4. Evaluation on the Image Quality Assessment Network

To evaluate the ability of Q-Net to assess image quality, we conducted an ablation study to quantify the performance contribution of Q-Net. Before that, this network was first evaluated and feasibility analysis was performed to generate a new dataset by adding different types and different intensities of synthetic noise to the acquired pCLE dataset images by noise addition process. We used three classical noise models: Gaussian noise, pretzel noise, and Poisson noise. The images with different noise intensities were scored separately, and the prediction scores of the datasets with different noises are shown in Figure 5, Figure 6 and Figure 7.

From this result, we can see that the predicted scores and the corresponding noisy images can be matched, and according to our visual subjective evaluation, it is obvious that the images with high noise intensity contain little information and possess lower scores. All three types of noise match this property, proving that the sub-IQA has good performance.

When applying IQA to the overall image denoising, we need the scores in the whole set of images and select the images with the highest scores to achieve the goal of improving image quality. We used two different pCLE images for the test, and the figure shows the score results for each image. It can be seen that the trend of the scores increases regularly and then decreases so that the image will be recovered best in a certain recovery process, as shown in Figure 8, and the figure also verifies this property. There is a certain difference in the optimal number of iterations for different images, which reflects that our need to add a quality assessment network is necessary; for example, if the number of iterations is set to 1200, image1 can achieve better results, but the other is not the best for recovery.

The above is the analysis of the Q-Net. For the next, we will conduct denoising experiments on the whole part to observe the effect.

### 3.5. Compare to Other Deep Learning Denoising Methods

We compared our method to three baselines: N2V, JOINT, which is the self-supervised blind denoising method that has shown the best results on the datasets we considered, and S2S, whose standard deviation is chosen to maximize the PSNR on the evaluation dataset. Figure 9 shows the effect of pCLE image denoising; we only compare the subjective visual quality due to the fact that pCLE does not exist in the real image. When evaluating the subjective quality and information content of an image, several factors come into play. Firstly, clarity and detail are paramount, with high-quality images exhibiting clear details and sharp edges. Additionally, color accuracy and contrast play crucial roles, as vibrant colors and appropriate contrast levels contribute to overall image quality. Furthermore, minimizing noise levels, particularly in darker areas and flat regions, is essential to maintain image integrity. Lastly, an image’s richness in conveying intended content or message is vital, requiring sufficient detail to effectively communicate its information.

It can be seen that there are still many foveal patterns in the Gaussian denoising method and, compared with other remaining methods, our method is better for the recovered image information, especially in the enlarged green area. For the last four sets of images in Figure 9, two regions of each image are zoomed in, and the red-boxed portion is the background, theoretically devoid of any information. The presence of excessive white pseudo-shadowing is evident in Figure 9c. The green bordered portion represents the magnification of the sample. It is clear that the contrast in part of Figure 9e is low, making it difficult to discern the location of the optical fiber. Although the contrast difference between Figure 9d,f is not significant, it is shown in subsequent experiments that our method provides more information and details than the JOINT method in “d”.

In order to provide a fair evaluation of image recovery, we also evaluate our method using objective metrics such as PSNR, SSIM, and also compare our method with four self-supervised denoising methods. We downloaded the fluorescence microscopy dataset W2S in the presence of real image and conducted comparison tests with each method on this dataset. Table 1 shows the metrics of all the results, and the PSNR and SSIM of our method are significantly better than the other methods in all three datasets.

Through subjective visual assessment, Figure 10, Figure 11 and Figure 12 demonstrate a notable denoising efficacy of our method on the W2S dataset.

These comparisons are important for demonstrating the efficacy of our denoising method and its superiority in terms of objective metrics and subjective visual quality.

The noise prediction network plays a role in the whole algorithmic architecture; it is able to provide more information to make up for the lack of information in D-Net itself and to provide more image features for subsequent training. We first compared the denoising effect of this algorithm with other algorithms, and, in order to verify the role of N-Net, we conducted an ablation experiment, which was used to quantify the role of N-Net. Table 2 demonstrates the comparison of metrics without and with the addition of N-net on the W2S dataset.

It can be seen that N-net extracts effective noise information in training and that this information has a positive effect on the denoising network.

## 4. Discussion

In this study, we compare the denoising effects of traditional denoising methods, self-supervised denoising methods, and jointly trained denoising methods on laser confocal microscopy imaging. We observe that, in contrast to other self-supervised denoising approaches, the incorporation of additional neural networks for assistance proves more beneficial in effectively extracting informative content from the images. Furthermore, within self-supervised denoising methods, neural networks, leveraging their inherent priors, demonstrate significant denoising capabilities. During the image restoration process, these networks first learn the relevant information within the image before addressing noise, resulting in an initial improvement followed by a deterioration in image quality. To select the most effective images during the recovery process, we employ a quality assessment network that we have trained, facilitating real-time evaluation of image restoration outcomes. On the other hand, we investigate the impact of joint training on the denoising network. Despite prior research in image noise prediction, the generalizability of such methods requires improvement. In this work, we consider the effectiveness of extracting noise information before engaging in joint training, and our results indicate not only superior image quality but also robust performance.

## 5. Conclusions

In this paper, we propose a self-supervised denoising algorithm to denoise images without clean images such as pCLE images. Current deep learning methods usually cannot be generalized to different samples, but our method achieves good results on fluorescence microscopy images as well. We exploited the structure of neural networks with a priori knowledge to verify that the trend of image recovery is to recover information before noise, and also designed a network to evaluate the results of image recovery by migration learning.

The combination of these three networks can solve the problem of denoising pCLE images and fluorescence microscopy images in complex environments. From the results, it can be seen that our method shows good results in terms of both subjective visual effects and quantitative metrics compared with the current state-of-the-art denoising algorithms.

## Figures and Tables

**Figure 1 sensors-24-02853-f001:**
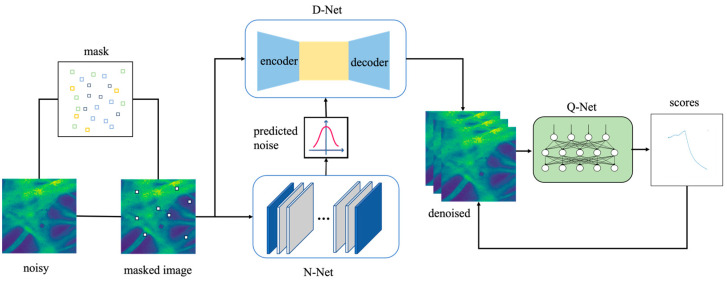
Architecture of proposed method.

**Figure 2 sensors-24-02853-f002:**
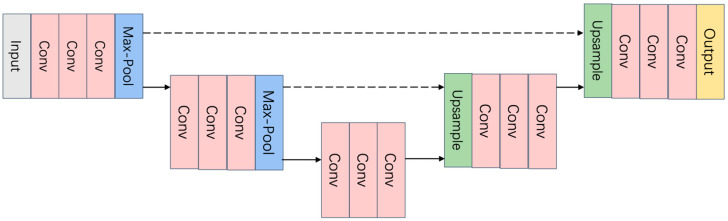
Architecture of the N-Net.

**Figure 3 sensors-24-02853-f003:**
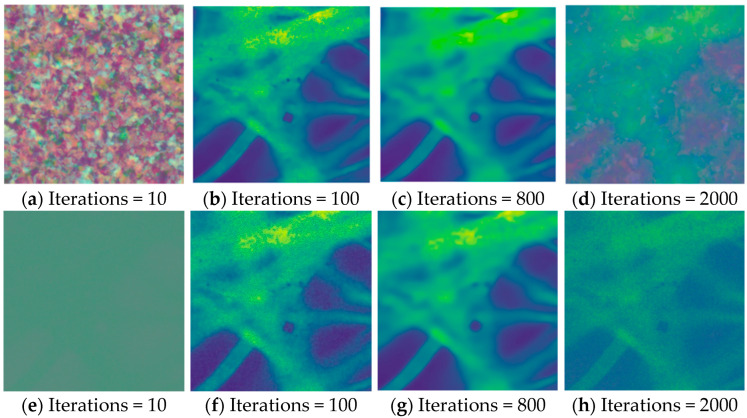
The reconstruction process of single image denoising. We show the reconstruction process of recovering a single image for DIP and S2S. Upper: DIP, Lower: Self2Self.

**Figure 4 sensors-24-02853-f004:**
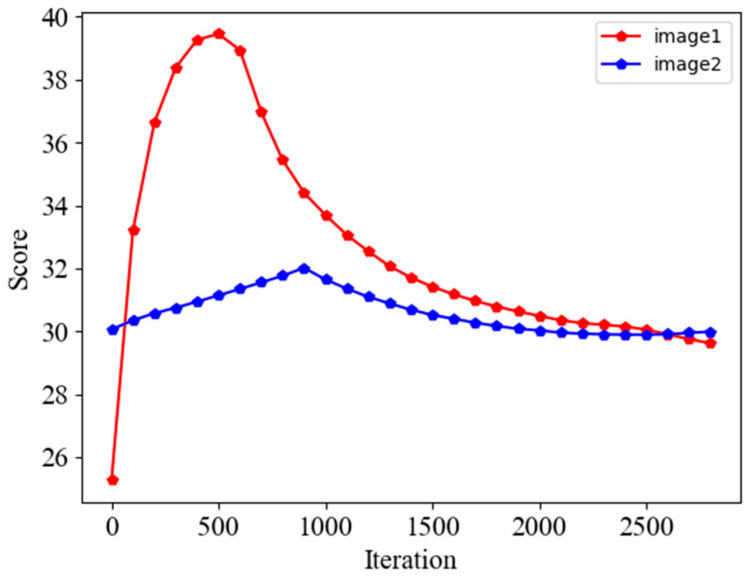
PSNR trends of the two datasets of W2S after passing through the denoising network.

**Figure 5 sensors-24-02853-f005:**
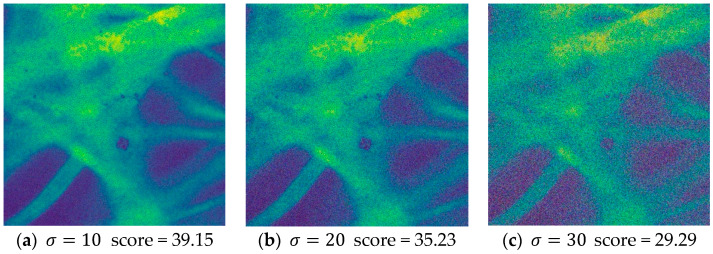
Quality assessment of the Gaussian noise images with different intensities.

**Figure 6 sensors-24-02853-f006:**
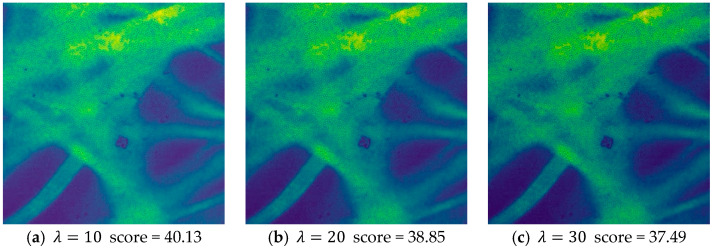
Quality assessment of the Poisson noise images with different intensities.

**Figure 7 sensors-24-02853-f007:**
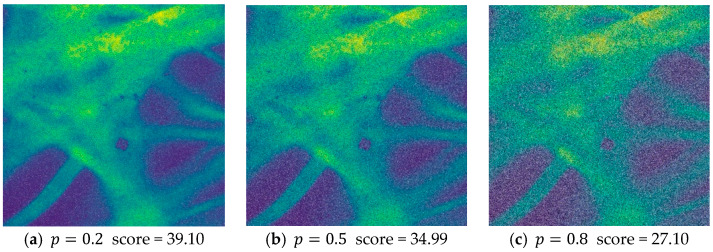
Quality assessment of the Pepper noise images with different intensities.

**Figure 8 sensors-24-02853-f008:**
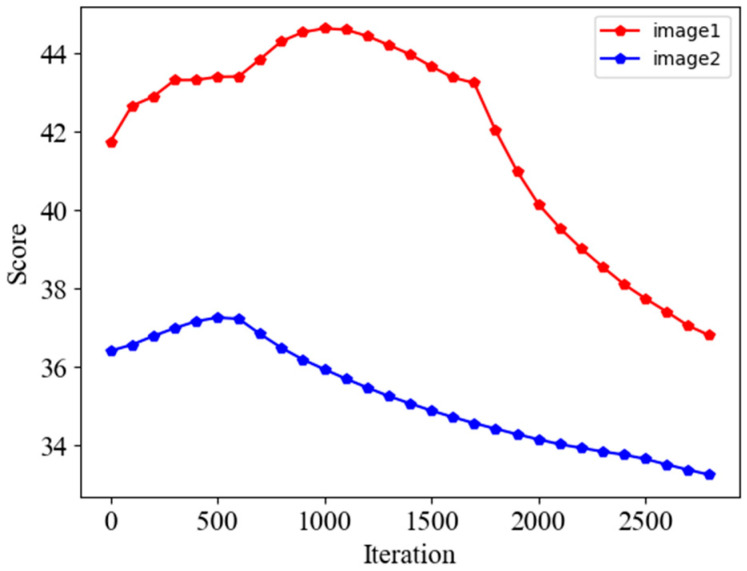
The trend of the scores of the two datasets of pCLE after passing through the two networks is increasing and then decreasing for all two images, and the number of iterations to reach the maximum score is different.

**Figure 9 sensors-24-02853-f009:**
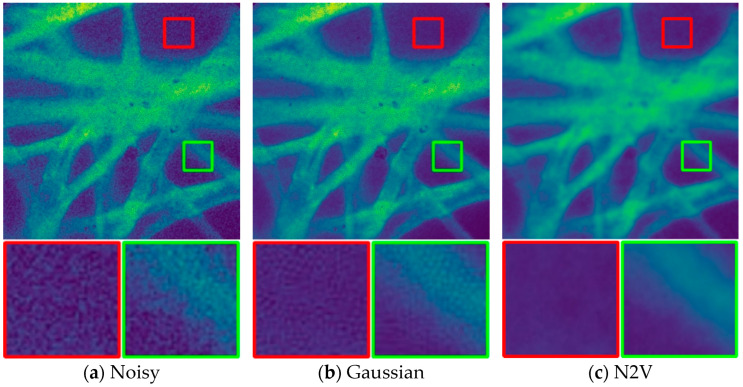
Denoising results of a pCLE image by different methods.

**Figure 10 sensors-24-02853-f010:**
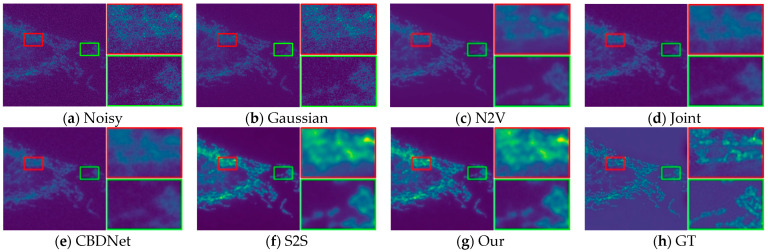
Visual comparison of denoising on the W2S-1 dataset.

**Figure 11 sensors-24-02853-f011:**
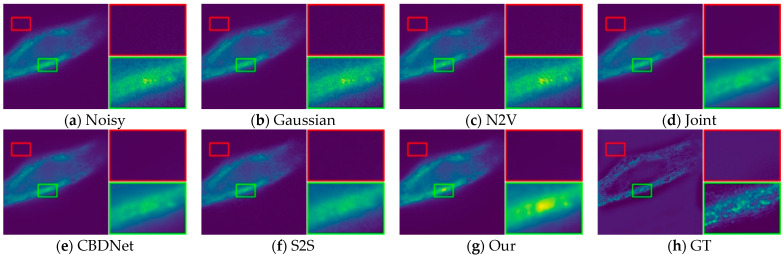
Visual comparison of denoising on the W2S-2 dataset.

**Figure 12 sensors-24-02853-f012:**
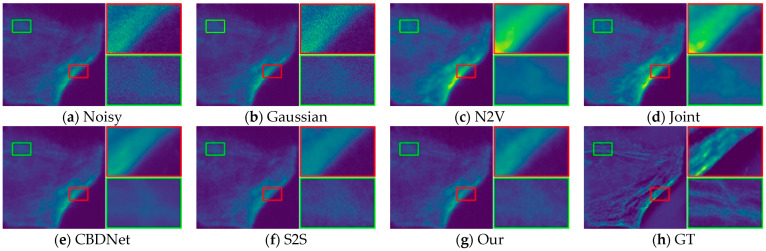
Visual comparison of denoising on the W2S-3 dataset.

**Table 1 sensors-24-02853-t001:** Evaluation of different methods on three datasets with PSNR and SSIM metric.

Algorithm	W2S-1	W2S-2	W2S-3
	PSNR/SSIM	PSNR/SSIM	PSNR/SSIM
NOISY	17.91/0.3432	15.54/0.2298	14.12/0.2151
GAUSSIAN	30.41/0.5176	30.55/0.4881	31.74/0.4295
N2V	32.73/0.8432	31.24/0.8546	33.50/0.8467
JOINT	32.73/0.8691	31.24/0.8678	33.50/0.8597
S2S	34.98/0.8664	33.22/0.8621	35.64/0.8732
OURS	35.67/0.8794	34.21/0.8721	36.14/0.8984

**Table 2 sensors-24-02853-t002:** Ablation study of the contribution of N-Net in terms of PSNR, SSIM.

Discriminator	W2S-1	W2S-2	W2S-3
	PSNR/SSIM	PSNR/SSIM	PSNR/SSIM
N-Net ✗	35.64/0.8631	33.79/0.8692	35.38/0.8781
N-Net ✓	35.67/0.8794	34.21/0.8712	36.14/0.8984

## Data Availability

Data are contained within the article.

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
