# Peer review of "Self-Supervised Joint Learning for pCLE Image Denoising†"

_sensors, 2024, doi:10.3390/s24092853_

Round 1

Reviewer 1 Report

Comments and Suggestions for Authors

The article introduces a novel self-supervised denoising method for addressing noise issues in probe-based confocal laser endomicroscopy (pCLE) images. This method combines a noise prediction network (N-Net), an image quality assessment network (Q-Net), and a denoising network (D-Net) to form a self-supervised framework that can be trained without paired clear-noise images. The approach leverages intrinsic prior knowledge in deep learning, learning image content first and then addressing noise during the image restoration process, thereby achieving effective denoising for both pCLE and fluorescence microscope images. Additionally, the method includes a reference-free image quality assessment metric, which is used to select the best denoised image during training, further improving denoising performance. Through experiments on the W2S dataset, the authors demonstrate that this method outperforms existing self-supervised denoising algorithms in terms of objective evaluation metrics and subjective visual quality. This method shows superior denoising effects on pCLE images and fluorescence microscope images compared to existing self-supervised denoising methods and holds significant practical value. The article still has the following issues:

1. 

In section 2.1, there are multiple instances of isolated closing parentheses.

2. 

In the last sentence of section 2.1, the paper mainly claims to explain N-Net, but it only provides a brief summary without specific analysis.

3. 

In section 3.4, when evaluating the capabilities of Q-Net, it would be beneficial to list the specific parameters for each type of noise.

4. 

There seems to be an error with figure 12(b) in section 3.5; the image appears to be misplaced.

5. 

The experimental results show that the proposed method slightly outperforms existing methods in objective evaluation metrics. However, from a subjective standpoint, there is no intuitive improvement, and the analysis and discussion of the experimental results in the paper are relatively brief. The comparative experiments are relatively singular, and it is necessary to add more comprehensive denoising effect comparison experiments with existing algorithms.

6. 

The paper sometimes refers to N-Net as a "noise prediction network" and at other times as a "noise estimation network." The author should standardize the terminology throughout the paper to ensure its professionalism and readability.

Overall, this paper presents a promising self-supervised denoising method with significant practical implications for the denoising of pCLE images. The methodology, experimental design, and results all indicate the effectiveness of this approach. However, it is recommended that the author revise the paper based on the suggestions provided and supplement additional comparative experiments and result analyses.

Comments on the Quality of English Language

Overall, the paper exhibits a high level of English writing proficiency suitable for publication in a scientific journal. The writing is clear and coherent, with a logical flow of ideas from the introduction to the conclusion. Each section builds upon the previous one, providing a comprehensive understanding of the research topic. But the paper some place refers to N-Net as a "noise prediction network" and at other places as a "noise estimation network." The author should standardize the terminology throughout the paper to ensure its professionalism and readability. 

Reviewer 2 Report

Comments and Suggestions for Authors

In this manuscript, the authors have presented a self-supervised denoising method, which integrates noise prediction networks, image quality assessment networks, and denoising networks in a collaborative, jointly trained manner. Superior results on pCLE images and fluorescence microscopy images can be obtained as described in this manuscript. But some issues need to be considered, as follows,

(1) In Introduction, excepting deep learning denoising methods, it suggested to introduce some traditional image denoising methods.

(2) In Results, some images and their processing results are provided, but the evaluations for the results are insufficient. So its suggested to add one or two more indexes to better evaluate your results.

(3) If other kinds of noise appear, such as impulse noise, is the proposed method still applicable?

(4) The "Discussion" is not sufficient and it seems a "Conclusion".  It is suggested to reorganized the "Discussion" or put it together with the "Results", i.e. "Results and Discussions".

Comments on the Quality of English Language

Readable.

Reviewer 3 Report

Comments and Suggestions for Authors

This paper focuses on denoising pCLE (and other W2S) images through self-supervised CNN, comparing the performance to other 3 algorithms: N2V, JOINT and S2S. The novelty of their approach lies in the implementation of an evaluation criterion to objetively determine the optimal number of iterations during image restoration, resulting in the improvement of the image quality.

Overall, the paper is well-writen and easy to follow, although some points could be considered to improve the clearness of the discussion and to reach a broader audience (not so familiarize with CNN arquitectures). The authors claim that the paper "mainly focuses on providing a comprehensive and detailed explanatation of the N-Net network". The quality assessment and final outcome of the denoising network are properly illustrated.  However, neither in the results nor in their discussion it is clear to me the contribution of the N-Net network. Further clarification on the actual influence of the N-Net would be highly appreciated.

In the following, some comments to the authors are listed:

- The figure 1 illustrate the "Arquitecture of the proposed method". It would be of interest to include the Q-Net step in the sketch to enrich the description of the entire pipeline. In addition, a first step related to the application of a mask to the noise image is included in the figure, which would be nice to comment in the text. Some information about this step of the Figure 1 is missing.

- In Figure 3, a recostruction process named DIP is used. No description or reference to this method is included in the whole text.

- When discussing Figure 4, it is said that PSNR of all images is calculated for 2,000 iterations. Does it mean up to iteration #2,000 or a total number of 2,000 measurements? The x range spans up to 2,700 roughly. In line with Figure 4, the figure captions says "three datasets of W2S", while just two plots are shown.

- The conclusion of Figure 5, 6, and 7 is that "it is obvius that the images with high noise intensity contain little information and possess lower score", so that IQA is doing a good job. However, to my eye, this is not the case in Figure 5, since (c) has a higher level of noise with a high score.

-Three images are mentioned in Figure caption 8, while just 2 plots are illustrated.

- The baselines used to compared are a bit confusing. First, at the beginning of point 3.5, N2V, JOINT and S2S baselines are described. Later, when going through the objective metrics to compare, these baselines are described again (a bit redundant), and, in addition, Figure 10, 11 and 12 include CBDNet algorithm, which is not even mentioned in the text.

Finally, I include a list of typos detected (to my understanding):

- Line 109: a couple of extra ")" in Q-Net and N-Net.

- Line 117: "(" right before sigma.

- Line 270: "." should be included instead of ",".

Round 2

Reviewer 2 Report

Comments and Suggestions for Authors

The authors has responded to my main concerns carefully.

Reviewer 3 Report

Comments and Suggestions for Authors

I appreciate the authors' efforts in addressing the concerns raised point-by-point. Based on their responses, I recommend that the manuscript be published in its present form.